# Evaluation of the Diagnostic Performance of a SARS-CoV-2 and Influenza A/B Combo Rapid Antigen Test in Respiratory Samples

**DOI:** 10.3390/diagnostics13050972

**Published:** 2023-03-03

**Authors:** Harika Öykü Dinç, Nuran Karabulut, Sema Alaçam, Hayriye Kırkoyun Uysal, Ferhat Osman Daşdemir, Mustafa Önel, Yeşim Tuyji Tok, Serhat Sirekbasan, Ali Agacfidan, Nesrin Gareayaghi, Hüseyin Çakan, Önder Yüksel Eryiğit, Bekir Kocazeybek

**Affiliations:** 1Department of Pharmaceutical Microbiology, Faculty of Pharmacy, Bezmialem Vakıf University, Istanbul 34093, Turkey; 2Department of Medical Virology, Başakşehir Çam and Sakura City Hospital, Istanbul 34480, Turkey; 3Department of Medical Microbiology, İstanbul Medical Faculty, Istanbul University, Istanbul 34093, Turkey; 4Department of Medical Microbiology, Cerrahpaşa Medical Faculty, Istanbul University-Cerrahpaşa, Istanbul 34098, Turkey; 5Department of Medical Laboratory, Eldivan Vocational School of Health Services, Techniques Çankırı Karatekin University, Çankırı 18100, Turkey; 6Blood Center İstanbul Şişli Hamidiye Etfal Training and Research Hospital, Istanbul 34360, Turkey; 7Department of Biology and Microbiology, Faculty of Arts and Sciences, Çanakkale Onsekiz Mart University, Canakkale 17100, Turkey; 8Health Vocational School Anesthesia, Istanbul Health and Technology University, Istanbul 34452, Turkey

**Keywords:** antigen tests, influenza A, influenza B, rapid test, SARS-CoV-2

## Abstract

This study aimed to evaluate the performance characteristics of a rapid antigen test developed to detect SARS-CoV-2 (COVID-19), influenza A virus (IAV), and influenza B virus (IBV) (flu) compared with those of the real-time reverse transcription-polymerase chain reaction (rRT-PCR) method. One hundred SARS-CoV-2, one hundred IAV, and twenty-four IBV patients whose diagnoses were confirmed by clinical and laboratory methods were included in the patient group. Seventy-six patients, who were negative for all respiratory tract viruses, were included as the control group. The Panbio™ COVID-19/Flu A&B Rapid Panel test kit was used in the assays. The sensitivity values of the kit were 97.5%, 97.9%, and 33.33% for SARS-CoV-2, IAV, and IBV, respectively, in samples with a viral load below 20 Ct values. The sensitivity values of the kit were 16.7%, 36.5%, and 11.11% for SARS-CoV-2, IAV, and IBV, respectively, in samples with a viral load above 20 Ct. The kit’s specificity was 100%. In conclusion, this kit demonstrated high sensitivity to SARS-CoV-2 and IAV for viral loads below 20 Ct values, but the sensitivity values were not compatible with PCR positivity for lower viral loads over 20 Ct values. Rapid antigen tests may be preferred as a routine screening tool in communal environments, especially in symptomatic individuals, when diagnosing SARS-CoV-2, IAV, and IBV with high caution.

## 1. Introduction

Severe acute respiratory syndrome coronavirus 2 (SARS-CoV-2) is a novel coronavirus which caused coronavirus disease 2019 (COVID-19). The SARS-CoV-2 virus was first detected in Wuhan, China, at the end of 2019, and it was quickly announced as a pandemic by the World Health Organization (WHO) [1]. The COVID-19 pandemic continues to have a major impact on healthcare and social systems worldwide [2]. Vaccination remains the most promising approach to controlling the COVID-19 pandemic [3]. However, because of the highly contagious nature of SARS-CoV-2, the lack of long-term immunity or a single, fully effective treatment against COVID-19 has resulted in a global pandemic, initiating a public health crisis that began in 2020 and remains active to this day [4].

Respiratory tract infections are important public health threats worldwide. Among pathogens, viruses including rhinoviruses, respiratory syncytial viruses, adenoviruses, influenza viruses, and parainfluenza viruses are responsible for most upper respiratory tract infections and some lower respiratory tract infections [5]. The SARS-CoV-2 which caused COVID-19 was recently added to the list of existing respiratory viruses. Influenza viruses and COVID-19 share very similar symptoms; however, the incubation period of SARS-CoV-2 is longer (2–14 days) than the flu caused by influenza viruses [6]. There is growing concern about the possibility of a simultaneous outbreak of SARS-CoV-2 and influenza viruses, especially in the winter season [7,8].

It can be difficult to distinguish COVID-19 from common viral infections based on clinical symptoms. Common viral infections exert non-specific clinical signs and symptoms, and it is important to differentiate COVID-19 from common viral infections to avoid misdiagnosis. A misdiagnosis may delay an accurate diagnosis and may result in further transmission throughout the community [9,10,11]. Since the clinical and epidemiological features of COVID-19 are similar to those of influenza, optimal management of these respiratory tract infections is crucial, as they are predicted to continue to circulate together [12,13]. Effective surveillance and diagnostic capacities must be provided, allowing us to monitor this and other respiratory viruses; this will form the basis of decisions regarding appropriate clinical management of the diseases involved [14]. Seasonal influenza (influenza A virus (IAV), and influenza B virus (IBV)), especially IAV, affects up to 10% of the adult population and 20% of children annually and displays substantial morbidity [15,16,17]. Early diagnosis of influenza viruses is critical, as current antiviral strategies are only effective in the early stages of the disease [18]. Therefore, differential diagnoses of SARS-CoV-2, IAV, and IBV are important, ensuring effective patient management and treatment.

Microbiologic diagnostic tests are used to differentiate between individuals with and without infectious diseases. Most infectious diseases have a “gold standard,” or benchmark test, against which alternative diagnostic tests can be assessed [19]. The two statistical criteria most frequently used to evaluate the performance of an alternative test relative to the gold standard are sensitivity and specificity [20]. Sensitivity is a test’s ability to correctly classify an individual as “diseased”. Specificity is a test’s ability to correctly classify a person as healthy. Sensitivity and specificity are inversely proportional, meaning that as the sensitivity increases, the specificity decreases [21]. A high sensitivity rate is vital when the test is used to identify a serious but treatable disease (e.g., COVID-19 or flu). The positive predictive value determines how many of the positive findings are true positives. The negative predictive value determines how many of the negative findings are true negatives. If the values rise toward 100, the test approaches the gold standard [21].

Molecular diagnostic tests based on the nucleic acid amplification test (NAAT) are the standard methods used to detect most viral respiratory tract infections [22,23]. According to the Centers for Disease Control and Prevention (CDC) and the WHO, the “gold standard” for clinical diagnostic detection of SARS-CoV-2 is laboratory-based NAATs [24,25]. The Infectious Diseases Society of America (IDSA) recommends rapid influenza molecular assays over rapid influenza diagnostic tests (RIDTs) for detecting influenza viruses in respiratory specimens of outpatients. The IDSA recommends using RT-PCR or other molecular assays to detect influenza viruses in respiratory specimens of hospitalized patients [26]. However, this technology can be relatively labor intensive and time consuming; laboratories often require specific infrastructure and trained staff to perform these tests [22]. This can become resource intensive; therefore, it may be beneficial to introduce automated or semi-automated molecular technologies that can be used at or near the point of care [27]. Many antigen-specific point-of-care (POC) test methods have been invented and used separately to detect SARS-CoV-2 and influenza A and B [28,29]. Although the CDC defines molecular tests for the detection of SARS-CoV2 and other respiratory infections such as IAV and IBV as the gold standard, it remains necessary to develop a multiplex POC device or rapid antigen tests for simultaneous early detection. Therefore, a POC kit or rapid antigen test that can detect multiple viruses from a single specimen using a single device would be very useful and would significantly decrease the turn-around time of the test. However, the WHO points out that the selection of tests should be based on proven performance (sensitivity and specificity) in the context of the intended use to optimize the testing strategy [26].

Although rapid, point-of-care molecular tests can shorten the diagnostic time, their use may be limited due to the high cost of these tests [22]. Many healthcare providers need a rapid test that is easy to use and inexpensive. Therefore, various rapid antigen tests have been developed to provide an alternative POC test with high sensitivity at a reduced cost [30,31]. Today, combo antigen tests have been developed for rapid diagnosis of SARS-CoV-2 and influenza cases. Although the sensitivity and specificity of these newly introduced tests have been determined by studies performed by the manufacturers, they may differ in routine practice and in their use among the general population. Therefore, we aimed to investigate the sensitivity, specificity, negative predictive value, positive predictive value, and kappa (κ) values of the SARS-CoV-2/IAV/IBV combo antigen test by detecting SARS-CoV-2 and IAV/IBV antigens qualitatively, using nasopharyngeal swab samples stored in appropriate conditions.

## 2. Materials and Methods

Nasopharyngeal and throat swab samples, taken from patients who applied to the relevant departments of different centers (Istanbul University-Cerrahpaşa; Başakşehir Çam, and Sakura Hospital, and Istanbul University) with clinical suspicion, were transferred to medical microbiology laboratories in viral nucleic acid buffer (VNAT) (BioNAT, Antalya, Turkey) for routine examinations. Samples that were positive for SARS-CoV-2, IAV, or IBV according to the rRT-PCR were included in our study as the samples of the patient group. We recorded their cycle of threshold (Ct) in which the viral load exceeded the detectable threshold. Archived samples were included as the control group; these samples were negative for common viral respiratory tract infections.

The sample sizes of the patient and control groups were calculated using the G Power V.3.1.9.4 analysis program. As a result of this calculation, the sample size of 100 SARS-CoV-2, 100 influenza A, and 24 influenza B positive cases (the study group), and 76 negative cases for viral respiratory tract viruses (the control group) was deemed acceptable. Viral nucleic acid extractions from the nasopharyngeal and throat swab samples with suspicion of IAV and/or IBV were performed on a Zybio EXM 3000 (Zybio, Shenzhen, China) device using the Rapid Nucleic Acid Extraction Kit (Bioeksen, Istanbul, Turkey). The detection of viruses from the extracted nucleic acids was performed using the Respiratory RT-qPCR MX-24S panel Kit (Bioeksen, Istanbul, Turkey) on a Bio-Rad CFX96 Touch instrument (Hercules, CA, USA). Double Gene RT-qPCR kit (Bio-speedy, Bioeksen, Istanbul, Turkey) on a Bio-Rad CFX96 Touch instrument (Hercules, CA, USA) was used to detect SARS-CoV-2.

The Panbio™ COVID-19/Flu A&B Rapid Panel (Nasopharyngeal) test is a rapid chromatographic immunoassay used for the qualitative detection of specific SARS-CoV-2, influenza A, and influenza B antigens present in human nasopharyngeal specimens, which is achieved using a single device. This lateral flow test (LFT) is based on immunochromatography and indicates the presence of the SARS-CoV-2 antigen with a colored line. The test contains a membrane strip precoated with antibodies specific to the nucleocapsid antigen of SARS-CoV-2, influenza A, and influenza B which was used to detect viruses. The Panbio™ COVID-19/Flu A&B Rapid Panel (Nasopharyngeal) test referred to as the combo test was performed according to the manufacturer’s instructions [32].

Before the research began, a total of 300 nasopharyngeal swab samples (100 SARS-CoV-2, 100 IAV, 24 IBV, 76 control) stored in a viral nucleic acid buffer (VNAT, BioNAT, Turkey) and under appropriate conditions were brought to room temperature. First, 300 µL of the vortexed sample was taken and mixed with the extraction buffer included in the test kit. It was vortexed for a few seconds to achieve homogenization. The nozzle cap was tight to the extraction buffer tube and four drops of the extracted sample were dispensed vertically into the sample well of the instrument. The results appeared as a band(s) of color after 15 min and were subsequently interpreted. All clinical specimens were studied in the Istanbul University-Cerrahpaşa, Cerrahpaşa Medical Faculty, Microbiology laboratory and Serology unit and the results were evaluated qualitatively. The test was conducted by two specialists who were blinded to avoid any observer bias. The results were interpreted as negative when only one band appeared on the C (control) line. The results were interpreted as positive when bands appeared on both the C and the tested lines (COVID-19, SARS-CoV-2; Flu A, influenza A; and Flu B, influenza B) (Figure 1). The test was interpreted as invalid if no band appeared or if the bands only appeared on the tested lines but not on the C line.

To evaluate the level of agreement between rRT-PCR and LFA, statistical evaluation was performed by accepting the rRT-PCR method as a gold standard. Statistical analysis was conducted using the IBM SPSS 20.0 (IBM Corp., Armonk, NY, USA) package program. Frequency (*n*), percentage (%), and mean values were determined. Sensitivity, specificity, positive predictive value (PPV), negative predictive value (NPV), likelihood, and accuracy values were calculated. Cohen’s kappa coefficient was used to assess the level of agreement between rRT-PCR and an antigen test; concordance was based on a value >0.6.

## 3. Results

The patient group was aged 1–81 and the control group was aged 1–76. There was no significant difference between the groups in terms of age and gender (*p* < 0.05). The sensitivities of LFA-based immunochromatographic card tests targeting SARS-CoV-2, IAV, and IBV antigens were evaluated in samples with viral loads ≤20 Ct (Ct range: 6–20) and >20 Ct (Ct range: 21–35).

When the Panbio™ COVID-19/Flu A&B Rapid Panel test was evaluated for SARS-CoV-2, the sensitivity value was 97.5% in samples with a viral load lower than 20 Ct. In samples with a viral load above 20 Ct values, the sensitivity value was 16.7%. When evaluated regardless of viral load, it was shown that the sensitivity, specificity, positive predictive value, and negative predictive values were 49%, 100%, 100%, and 60.3, respectively. When IAV cases were evaluated, the sensitivity value was 97.9% in samples with a viral load of lower than 20 Ct, and 36.5% in samples with a viral load above 20 Ct. When evaluated regardless of viral load, it was shown that the sensitivity, specificity, positive predictive value, and negative predictive values were 66%, 100%, 100% and 69.1, respectively. When IBV cases were evaluated, the sensitivity was 33.3% in samples with a viral load of below 20 Ct, while the sensitivity was 11.1% in values above 20 Ct. When evaluation was carried out regardless of viral load, the sensitivity, specificity, positive predictive value, and negative predictive values were 25%, 100%, 100%, and 80.9, respectively. The sensitivity, specificity, PPV, NPV, and kappa values of the kit tested are given in Table 1. The positivity and negativity levels according to the Ct values of SARS-CoV-2, IAV, and IBV cases are shown in Figure 2, Figure 3 and Figure 4.

In the analysis performed to evaluate the level of agreement between rRT-PCR, which is the gold standard method in the diagnosis of COVID-19, and the rapid antigen test used, Cohen’s kappa coefficient was 0.98 for the Ct below 20. For the diagnosis of IAV and IBV, Cohen’s kappa coefficient was 0.98 and 0.46 for the Ct below 20, respectively (Table 1). When Ct values were evaluated according to ROC curve analysis, significant results were obtained for SARS-CoV-2, IAV, and IBV at 20, 22, and 15 Ct values, respectively (p < 0.05). The relationship between sensitivity and specificity is given in Table 2 based on ROC curve analysis.

## 4. Discussion

In our study, we performed a sensitivity analysis using the Panbio™ COVID-19/Flu A&B test, which is an LFA-based rapid antigen test and compared the performance of the kit with the rRT-PCR method used to detect SARS-CoV-2, IAV, and IBV. Sensitivity below 20 Ct was 97.5%, 97.9%, and 33.3% for SARS-CoV-2, IAV, and IBV, respectively. Above 20 Ct, the sensitivity was 16.7%, 36.5%, and 11.1% for SARS-CoV-2, IAV, and IBV, respectively. The specificity of the test was 100% for all viruses. Ct values are inverse to the viral RNA copy numbers; therefore, a lower Ct value indicates a high viral load [33]. However, studies have reported that the time from symptom onset and Ct values affect the sensitivity of rapid antigen tests used to detect the presence of SARS-CoV-2 in nasopharyngeal samples [34]. This study demonstrated that the sensitivity of the rapid antigen test is higher at a lower Ct value.

Several factors that may affect the sensitivity of the rapid antigen tests include methodology differences, the severity of the infection, and the sample types. Most of the studies included stored specimens and had no information related to the time from symptom onset. Oh et al. [35] evaluated the sensitivity of the STANARD Q COVID-19 Ag test with RT-PCR in diagnosing COVID-19 and concluded that the differences mainly originated from different methods of RT-PCR. The Ct values were not comparable between the RT-PCR tests. Lee et al. [34] made sure to use fresh swab specimens and correctly recorded the time from symptom onset values. The sensitivity of rapid antigen tests is known to decline when using stored specimens, as seen in the study of Parvu et al. [36]. The sensitivity of the rapid antigen test in their study declined from 75.3% (in fresh specimens) to 70.9% (in frozen specimens). Meanwhile, Igloi et al. [37] reported that the sensitivity of their Q Ag rapid antigen test for COVID-19 increased for specimens collected within 7 days of symptom onset and the sensitivity of the Q Ag rapid antigen test increased to 99.1% when the Ct value of the E gene was <25. In a similar study conducted by Kim et al. [38], the Ag test sensitivity increased with Ct ≤ 30 and for specimens collected 1 to 5 days post-symptom onset. They suggested that Ct values of rapid-antigen-test-positive specimens may not accurately indicate patient status, while Ct values may vary with the specimen quality and may not correlate with the presence of SARS-CoV-2 antigens.

Therefore, the performance of the rapid antigen test was evaluated by grouping the samples that were found to be positive for SARS-CoV-2, IAV, and IBV in the symptomatic period by rRT-PCR as ≤20 Ct and >20 Ct. For SARS-CoV-2 and IAV, the sensitivity of the rapid antigen test was 97.5% and 97.9%, respectively, when the Ct value was below 20, and a very significant decrease in sensitivity was detected when the Ct value was above 20. For IBV, the same result with SARS-CoV-2 and IAV was obtained when the kit’s sensitivity was evaluated according to the Ct values. However, despite the low Ct value in the rapid antigen test, a very low sensitivity rate was found for the diagnosis of IBV. When the compatibility between the Panbio™ COVID-19/Flu A&B rapid panel test and rRT-PCR for the detection of SARS-CoV-2, IAV, and IBV viruses was evaluated, different results were obtained by the viruses. SARS-CoV-2 and IAV showed strong coherence with Cohen’s kappa value (0.98), which was an almost perfect match. In contrast, IBV showed moderate coherence. The rapid antigen test used in the present study may be preferred as an alternative to rRT-PCR in the early phase diagnosis of SARS-CoV-2 and IAV since it has a sensitivity of over 97% and a specificity of 100%.

In a study conducted by Widyasari et al. [39] in which the performance of a COVID/FLU combo antigen test was compared to an rRT-PCR SARS-CoV-2, IAV, and IBV combo antigen test, the sensitivity values were 93.1%, 92.2%, and 91.18%, respectively. The researchers used a STANDARD Q COVID/FLU Ag Combo test (Korea), a rapid chromatographic immunoassay used for the qualitative detection of specific SARS-CoV-2, influenza A, and influenza B antigens present in human nasopharyngeal specimens. The test is performed using a single device. They reported a Cohen’s kappa index value of 0.940 and a Cohen’s kappa value of 0.941 and 0.928 for influenza A and influenza B, respectively. These values indicated substantial agreement between the Ag Combo test and rRT-PCR. The researchers also restricted the duration from symptom onset and the Ct value of *RdRp* for SARS-CoV-2 and analyzed the sensitivity of the Q Antigen combo test used to detect the presence of SARS-CoV-2 in the samples according to the different durations from symptom onset and Ct values. The sensitivity of the Q Ag combo test reached up to 100% within a week (0–6 days). However, when used to assess samples collected at the duration from symptom onset > 7 days, the sensitivity of the Q Antigen combo test decreased significantly. When the sensitivity values of samples with Ct values of *RdRp* ≤ 20 were evaluated, the sensitivity of the combo antigen test was higher than the samples with Ct values between 20 and 30. They concluded that the Q Antigen combo test has a very high sensitivity and specificity for the detection of SARS-CoV-2, influenza A, and influenza B in a single sample with a single device. They considered the Q Antigen combo test a considerably useful tool for the detection and differentiation of SARS-CoV-2, influenza A, and influenza B, providing benefits such as cost-effectiveness, easy handling, and the ability to detect multiple viruses using a single device with a short turnaround time [39]. In addition, it has been shown that symptom onset is also effective in the diagnostic performance of rapid antigen tests; the sensitivity of the Q Antigen combo test was 100% when the samples were collected within one week (0–6 days). As expected, when the specimens were collected at >7 days, the sensitivity of the Q Antigen combo test decreased significantly [39]. Similarly, the sensitivity, specificity, positive predictive, and negative predictive values of a different combo antigen test (newly developed antigen test QuickNavi-Flu+COVID-19 Antigen test) for the detection of SARS-CoV-2 from nasopharyngeal samples were 80.9%, 99.8%, 98.7%, and 95.8%, respectively. The sensitivity reached 88.3% in symptomatic cases. However, the fact that the sensitivity was over 95% for Ct values below 20 regardless of symptoms, and for Ct values 25–29, the sensitivity decreased to 46.2%. The sensitivity of their kit decreased with increasing Ct values. For Ct values ≥ 30, the sensitivity also decreased to 25.0% in asymptomatic cases. The researchers concluded that the QuickNavi-Flu+COVID19 Antigen test indicated a desirable sensitivity and specificity for SARS-CoV-2 detection using both nasopharyngeal and anterior nasal samples, especially in symptomatic patients. The sensitivity, specificity, positive predictive values, and negative predictive values of the studied kit were compatible with ≤20 Ct results, except those relating to influenza B. The sensitivity values differed between symptomatic and asymptomatic cases in the 25–29 Ct value range. For the studied nasopharyngeal and anterior nasal specimens, the median Ct values were lower for the symptomatic cases compared to the asymptomatic cases. This may have been caused by the difference in sensitivities between the symptomatic and the asymptomatic cases in this Ct range [40]. In our previous study, SARS-CoV-2 RNA-positive respiratory tract samples with viral loads of <25 Ct (cycle of threshold), 25–29 Ct, 30–35 Ct, and <35 Ct, a total of 205 patient samples were studied by the lateral flow method using twelve commercial rapid antigen tests from different companies, and their performance was evaluated. We also reported that the sensitivities of the kits decreased in proportion to the increase in Ct values [41]. Therefore, the data obtained from different studies are compatible with our study, except for that of influenza B. As a result of the Roc curve analysis of this study, we showed that the influenza B test is more sensitive when detecting patients with a Ct value of 15 and below (*p* = 0.03). For SARS-CoV-2 and IAV, we also showed that it was more sensitive when detecting patients with a Ct of 20–22, which is consistent with other study results (*p* < 0.00, *p*: 0.02, respectively). Therefore, it should be considered that negative antigen results may be associated with viral load in patients with suspected symptoms.

COVID-19 and influenza have similar symptoms, and these similarities make differential diagnosis very difficult. The Panbio™ COVID-19/Flu A&B Rapid test was developed to detect SARS-CoV-2, influenza A, and influenza B using nasal or oropharyngeal swabs and the total test time is about 15–20 min. Thus, it is possible to identify infected individuals very early and to take precautions to prevent the spread of the three viruses detected by this combo rapid antigen test. Firstly, it is important to understand the meaning of Ct values. The Ct values show the number of amplification cycles required for the target gene to exceed a threshold level in an rRT-PCR assay [42]. The Ct values are correlated with SARS-CoV-2 accumulation and the clinical presentation of patients. These values may be regarded as a surrogate for the determination of viral load [43,44].

Our study has several limitations. We included patients who presented to the hospital with clinical symptoms, but patients who were not clinically suspected to be referred to hospitals were excluded from the study. The patients were not recruited during the COVID-19 outbreak, which led to a drastic decline in influenza transmission, and there may be clinical differences in symptoms of patients with COVID-19. Additionally, laboratory parameters may be different due to the predominant strains of the SARS-CoV-2 virus and influenza virus strains circulating at different time points. Another limitation of our study is the small number of patients diagnosed with influenza B among the included cases. Sample insufficiency for this virus might have negatively affected the sensitivity of the test. However, one of the factors affecting sensitivity might be the antigenic target in the LFA-based test. Although SARS-CoV-2, IAV, and IBV share many symptoms, they are also highly contagious. Since the differential diagnosis of these viruses is difficult due to non-specific symptoms, it is necessary to develop clinically validated LFA-based antigen tests with high sensitivity and specificity rates that enable the differentiation of SARS-CoV-2 and influenza viruses in a single test. Antigen tests are advantageous compared to molecular methods in that they give faster results, are easier to use, and have lower costs. On the other hand, we used specimens of patients that were positive for SARS-CoV-2, IAV, and IBV in the symptomatic period. The specificity of the Panbio™ COVID-19/Flu A&B test was 100% for SARS-CoV-2, influenza A, and B, as reported in many similar studies [34,45]. A diagnostic test with very high specificity will rule out healthy individuals and will also eliminate false-positive results. This means additional tests will not be used for false-positive results.

In conclusion, this kit demonstrated high sensitivity to SARS-CoV-2 and IAV for viral loads below 20 Ct values, but the sensitivity values were not compatible with PCR positivity for lower viral loads over 20 Ct values. However, the results of this test should be approached with extreme caution because the Panbio™ COVID-19/Flu A&B Rapid Panel test kit is prone to produce false negatives for the higher Ct values in response to low viral loads during the detection of SARS-CoV-2, INF-A, and INF-B. Rapid antigen tests may be preferred as a routine screening tool in communal environments, especially in symptomatic individuals, when diagnosing SARS-CoV-2, IAV, and IBV with high caution. There is a clear correlation between lower Ct values and the presence of clinical signs, which is especially evident in symptomatic patients, but the diagnostic value of these rapid antigen tests will remain controversial unless their sensitivity reaches a satisfactory level for non-symptomatic patients with high Ct values in rRT-PCR. However, it is very important to perform these tests in the first days of symptom onset (the early stage) when the viral load is high.

## Figures and Tables

**Figure 1 diagnostics-13-00972-f001:**
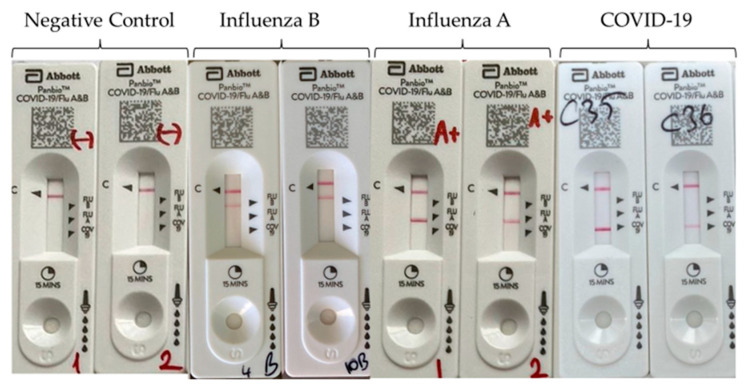
The Panbio™ COVID-19/Flu A&B Rapid Panel (nasopharyngeal) test results. The control line (C) begins to appear around 3–4 min following the application of the sample–buffer mixture on the device. The other line will also appear next to the test lines when the samples contain antigens of influenza B (Flu B line), influenza A (Flu A line), or SARS-CoV-2 (COVID-19 line). One line on the C marker indicates that the test is negative. Two lines—one on C and one on either Flu-B, Flu-A, or COVID-19 markers—indicate that the test is positive either for SARS-CoV-2, influenza A, or influenza B.

**Figure 2 diagnostics-13-00972-f002:**
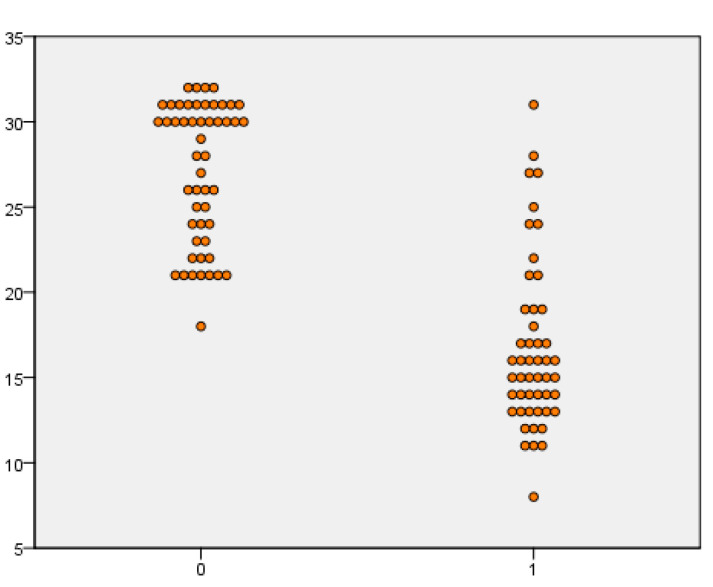
Comparison of SARS-CoV-2 rapid antigen test results in rRT-PCR Ct values.

**Figure 3 diagnostics-13-00972-f003:**
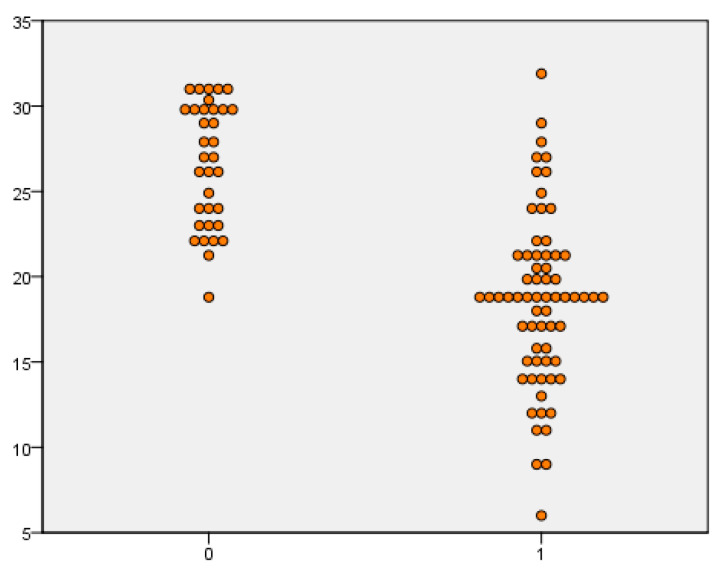
Comparison of influenza A rapid antigen test results in rRT-PCR Ct values.

**Figure 4 diagnostics-13-00972-f004:**
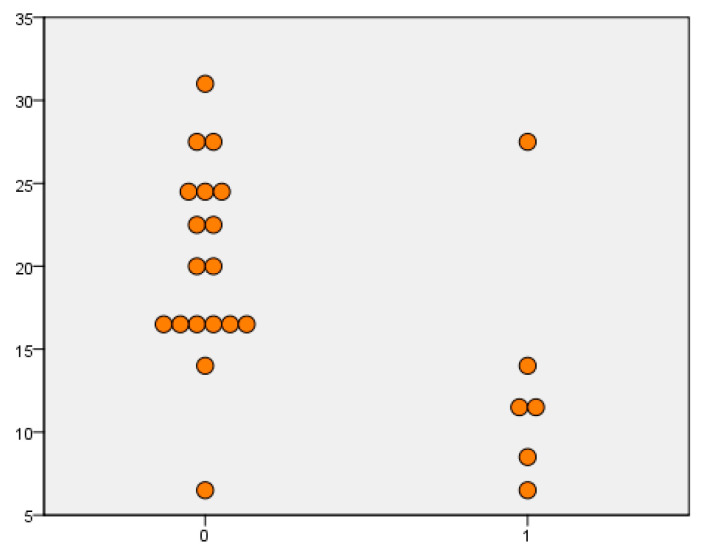
Comparison of influenza B rapid antigen test results in rRT-PCR Ct values.

**Table 1 diagnostics-13-00972-t001:** Evaluation of the diagnostic performance of lateral flow tests study on nasopharyngeal swab samples of patient groups with viral loads of ≤20 Ct and >20 Ct diagnosed with SARS-CoV-2, IAV, and IBV via rRT-PCR.

			Sensitivity (%)	Specificity (%)	PPV (%)	NPV (%)	Kappa
SARS-CoV-2	≤20 Ct	(*n* = 40)	97.5	100	100	98.7	0.98
	>20 Ct	(*n* = 60)	16.7	100	100	60.3	0.18
	Total	(*n* = 100)	49	100	100	59.8	0.45
IAV	≤20 Ct	(*n* = 42)	97.9	100	100	98.7	0.98
	>20 Ct	(*n* = 58)	36.5	100	100	69.7	0.41
	Total	(*n* = 100)	66	100	100	69.1	0.63
IBV	≤20 Ct	(*n* = 15)	33.3	100	100	88.4	0.46
	>20 Ct	(*n* = 9)	11.1	100	100	90.5	0.18
	Total	(*n* = 24)	25	100	100	80.9	0.34

PPV: Positive predictive value, NPV: negative predictive value, Ct: cycle threshold, and INF: influenza.

**Table 2 diagnostics-13-00972-t002:** Evaluation of diagnostic performance according to Ct values of tests including ROC curve analysis.

ROC Curve Parameters			
	SARS-CoV-2	Influenza A	Influenza B
AUC	0.928	0.907	0.801
95% CI (min-max)	0.877–0.980	0.849–0.94	0.522–1
*p*	<0.001	0.029	0.03
Cut-off	20	22	15
Sensitivity (%)	79.6	81.8	83.3
Specificity (%)	98	94.1	88.9

AUC: Area under the curve and CI: confidence interval.

## Data Availability

Data of this study are available from the corresponding author upon request.

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
