# Peer review of "Evaluation of the Diagnostic Performance of a SARS-CoV-2 and Influenza A/B Combo Rapid Antigen Test in Respiratory Samples"

_diagnostics, 2023, doi:10.3390/diagnostics13050972_

Round 1

Reviewer 1 Report

Performance characteristics of the rapid antigen test developed for the detection of SARS-CoV-2, INF-A, and INF-B in the diagnosis of COVID-19 and flu cases from naso-pharyngeal samples and compared to gold standard method rRT-PCR. PCR confirmed positive and negative clinical samples were screened with the Panbio™ COVID-19/Flu A&B Rapid Panel test, an lateral flow strip immunoassay for nucleocapsid proteins of CoV-1, Influenza A and B. The results were evaluated based on PCR ct values. Some minor issues are listed as follows.

1.       Please check the grammar for the following sentences.

Lines 39-41: “COVID-19 is caused by the SARS-CoV-2 virus.....”

Line 46 “..a global pandemic and public health crisis that began in 2020 but continues today“

Line69: “tory tract infections, which are predicted to continue to circulate together”

Line 81: “A/B is recently critical for effective patient management and treatment”

And for meaning:

Lines 89-91: “A POC kit that can be detect multiple viruses from a single specimen using a single 89 device would be very useful and time-consuming by significantly decreasing the test turn- 90 around time.”

2.       Lines 135-136: I don’t understand this sentence “Following sample collection, a swab in the buffer tube was moved around in a circle in place and squeezed into the tube wall.”

3.       What was the sensitivity, specificity etc values for the commercial LFA test? These can be included in the disscussion section.

Author Response

Thank you for your valuable contribution and suggestions. The answers are listed below.

  1.  Grammar edited. In addition, an editing service was provided for the manuscript.
  2. We have removed the sentence on lines 135-136. The test procedure has been given more clearly and in detail on lines 150-155.
  3. In line with the reviewer's suggestions, the sensitivity, specificity, etc. values of the commercial LFA test were added to the first paragraph of the discussion.

Reviewer 2 Report

The information provided in the manuscript is interesting. However, I have a couple of suggestions for improving the paper:

Introduction. This part is quite detailed. It should be more specific to the topic.

Methods and Materials. Тhere should be clearly defined sub-points of the individual materials and methods. It is very narratively presented. It is unclear how many samples were tested.

Discussion. The first paragraph is not appropriate. The paragraph between lines 317-337 is very detailed. In general,  the discussion contains a lot of information about other articles and studies, these parts may be shortened.

Author Response

Thank you for your valuable contribution and suggestions. The answers are listed below.

  1. In line with the reviewer's suggestions, we revised the introduction part more specifically to the topic.
  2. The number of patients included in the study has been given in the material-method section in lines 153-158, and all of these patients were tested. The test procedure is explained in more detail.
  3. It has been revised in the first paragraph of the discussion. In the first paragraph, the data of the study are presented.

Reviewer 3 Report

“Evaluation of Diagnostic Performance of SARS-CoV-2 and Influenza A/B Combo Rapid Antigen Test in Respiratory Samples” by Dinc et al. is an interesting study. This reviewer has carefully read the manuscript and provided the following suggestions to the authors to consider. 

1.      “SARS-CoV2 virus” should be corrected to read “SARS-CoV2” throughout the text.

2.      Line 44: "contagiousness" should be written contagious.

3.      Lines 47-50: Provide references for this sentence.

4.      Lines 50-51: "severe acute respiratory syndrome coronavirus 2 (SARS-CoV2)" is spelled out here. It should be spelled out in the first paragraph at the first use.

5.      Lines 54-55: Provide reference(s).

6.      Line 56: "coronavirus disease 2019 (COVID-19)". Be consistent in using the terminology; it should be spelled out at the first use in first paragraph, not here.

7.      Lines 59-60: "A misdiagnosis may cause a delayed diagnosis and may result in further transmission in the community [6]". Provide more references to support this.

8."Overall, seasonal influenza, especially INF-A, affects up to 10% of the adult population and 20% of children annually and displays substantial morbidity [10]". Provide more references. Also, INF-A is not a standard acronym for influenza A virus, rather prefer using IAV.

9.      Line 78: stretching. Do you mean relaxing?

10.   Line 81: "recently critical"?

11.   Lines 90-91: “would be very useful and time-consuming by significantly decreasing the test turn-around time”. This is contrasting statement. I believe authors intended to write “time saving”.

12.   Lines 82-91: This reviewer has a different perspective regarding these statements made by the authors. Authors are trying to imply that current molecular detection methods of respiratory virus infections are cumbersome and time-consuming and need certain trained specialist to perform, therefore, a multiplex point-of-care (POC) method for antigen-detection would be a better option. This is how the authors are trying to justify the need of a multiplex POC method/device for rapid detection. This reviewer would suggest the authors explain what molecular detection methods are recommended by the WHO and CDC for respiratory viruses such as SARS-CoV2, IAV and IBV. And, if these recommended tests offer higher sensitivity and specificity over other available methods, and why? A brief information on the available point-of-care methods or devices and rapid antigen tests, with specificity and sensitivity. The authors must emphasize that while WHO or CDC recommended molecular tests are the gold standards for the detection of SARS-CoV2 and other respiratory infections such as IAV and IBV, there remains a need of developing a multiplex point-of-care device or rapid antigen tests for simultaneous detection of these three virus pathogens, for early detection of these viruses, preferably at the initial stage by the user at home, with objective to limit the exposure to others and further transmission.

13.   Lines 168-170: “When the Panbio™ COVID-19/Flu A&B Rapid Panel test was evaluated for SARS-COV-2, the sensitivity value was found to be 97.5% in samples with a viral load lower than 20 Ct.” The authors did not provide the data on the limit of detection (LOD). Ct higher than 20 still represents a very broad range of molecular detection (up to Ct 38, at least). Up to what upper Ct value was the developed LFA test was able to detect the three viruses? For example, up to Ct 35, or 38?

14.   Line 170: “In samples with a viral load above 20 Ct values, the sensitivity value was 16.7%.” Samples with Ct values between 20 and 30 would have significant viral loads for the viruses under investigation. A 16.7% sensitivity appears to be quite low, at least in this range of Ct (Ct 20 to 30). The authors did not mention if they had samples with higher Ct values, such as above 30 or 35 Ct, that were negative for the LFA test developed.

15.   Table 1: While samples with Ct less than 20 had high sensitivity for SARS-CoV2 and IAV detection, the sensitivity for IBV was poor (33.3%). In addition, samples with Ct higher than 20 had consistently poor sensitivities for all the three virus pathogens. These data reflect that the developed assay is prone to produce false negatives for the higher Ct values, in case of low viral loads, for the virus pathogen(s) under investigation.

16.   This reviewer strongly believes that the authors must have included at least one more target Ct value (for example, Ct 30 or beyond) to verify if the developed assay is able to detect the viruses in that range? Unfortunately, there is no information on the limit of detection of the assay.

17.   Line 208: “in human tissue fluids”?

18.   Lines 254-258: “The US Food and Drug Administration (FDA) emphasizes that a rapid antigen test should have at least 80% sensitivity and 98% specificity [30]. So, the rapid antigen test used in the present study can be preferred as an alternative to rRT-PCR in the diagnosis of SARS-CoV-2 and INF-A, since it has a sensitivity of over 97% and a specificity of 100%.” The provided reference is misleading and incorrect; this is a news article on UW website and does not provide the details on the FDA regulations, as claimed by the authors. Can FDA accept false negative results when qRT-PCR Ct value is above 20 in a given sample, for SARS-CoV2 and IAV or IBV? Please provide specific regulations with correct citation(s) and how would it impact the present study.

19.   Lines 259-282: The authors mentioned a recent study by Widyasari et al. 2023, published in MDPI-Diagnostics, and tried to justify their rate of sensitivity for SARS-CoV2 and IAV detection with previously reported assay. While Widyasari et al. 2023 reported high sensitivity of SARS-CoV2 detection, using RdRp, in samples with Ct values between 20 and 30 as well as 30 and 40, the current study presented here does not have any data in these Ct ranges. Rather the authors chose a very broad Ct range (20 and above). Therefore, they unfortunately cannot compare the sensitivity of the developed assay with the one reported by Widyasari et al. 2023.

20.   This reviewer suggests that the authors make necessary changes to the manuscript and discuss the limitations of their study with future recommendations for improvement of such assays. Introduction and Discussion sections require significant improvements. Some references need to be corrected. 

21. Abstract: "the sensitivity values were shown to be compatible with PCR positivity." Enough data not provide to support this statement. 

22.   The manuscript also needs significant English language corrections, for better reading. This reviewer suggests authors take help of a native English language speaker to proof-read and correct the language throughout the text.

Author Response

Thank you for your valuable contribution and suggestions. The answers are listed below.

Best regards,

  1. It was corrected throughout the text.
  2. It has been edited.
  3. We have added the new reference (R-5) to lines 49-50.
  4. Abbreviations have been made in the first paragraph.
  5. We have added the new references (R-7,8) to lines 56-57.
  6. Abbreviations have been made in the first paragraph.
  7. We have added the new references (R-10, 11) to lines 61-62.
  8. The abbreviations for Influenza A and Influenza B have been changed to IAV and IBV. The following references were added (R-16, 17).
  9. The sentence on line 78 was removed after the revision regarding the shortening of the introduction of the other reviewers.
  10. It has been edited. Replaced "recently critical" with "important"
  11. Thank you for your attention. As you mentioned, we revised it as “time-saving “.
  12. The molecular detection methods that are recommended by the WHO and CDC for respiratory viruses such as SARS-CoV2, IAV, and IBV were added between lines 86-106 as “According to the Centers for Disease Control and Prevention (CDC) and WHO, the “gold standard” for clinical diagnostic detection of SARS-CoV-2 is laboratory-based NAATs [24, 25]. The Infectious Diseases Society of America (IDSA) recommends rapid influenza molecular assays over rapid influenza diagnostic tests (RIDTs) for detecting influenza viruses in respiratory specimens of outpatients. IDSA recommends using RT-PCR or other molecular assays to detect influenza viruses in respiratory specimens of hospitalized patients [26].”. These recommended tests offer higher sensitivity and specificity over other available methods but these methods require expensive equipment and skilled workers. Rapid antigen tests offer a short time to have a result and higher sensitivity and specificity but sometimes this goal is not realistic especially for specimens with low viral loads. We totally agree with the reviewer and we also presented shortcomings of multiplex point-of-care tests and they are not effective as recommended molecular tests, until yet. We also added to the last paragraph of the discussion the following sentence “However, the results of this test should be approached with extreme caution because the Panbio™ COVID-19/Flu A&B Rapid Panel test kit is prone to produce false negatives for the higher Ct values in response to low viral loads during the detection of SARS-CoV-2, INF-A, and INF-B.”.

  13. The manufacturer recommends that the Panbio™ COVID-19/Flu A&B Rapid Panel test can achieve the most accurate result in clinical samples in the 15-20 Ct range.  The number of our patient specimens for SARS-CoV-2 was           ≤20 Ct  (n= 40) and >20 Ct (n= 60), for IAV, was ≤20 Ct (n= 42) >20 Ct (n= 58) and for IBV was  ≤20 Ct (n= 15) 33.3 >20 Ct (n= 9) 11.1. As it can be seen 20 Ct seems to be a medium value for all three viruses. The over 20 Ct group will probably give more false negative results because there are only 16 specimens over 30 Ct for the INFA group and only one for the INFB group and 5 for the SARS-CoV-2 group. 20 Ct seems very optimal regarding to the recommendation of the producer and also looking at our detected values. The over 20 Ct group also covers the 20-35 Ct range and we suggest that the results obtained in all three viruses in this group are not sufficient to have reasonable/optimal sensitivity and specificity.

  14. In the manufacturer's insert, it is stated that the sensitivity is higher in samples with Ct values ​​of ≤30 for SARS-CoV-2, and with a considerably lower sensitivity of 30.0% in samples with Ct values ​​of >30. For the other two viruses, no data were shared in terms of Ct values by the manufacturer. As we mentioned in the first submitted article, we determined the grouping according to the Ct range with an objective approach within the possibilities at hand.

    We can share the Ct values ​​of the samples included in our study with the referee. For SARS-CoV-2, IAV and IBV with Ct values ​​in the range of 30-35, 26 (25 negatives, 1 positive for LFA test), 22 (21 negatives, 1 positive for LFA test), 1 (negative for LFA TEST) is. We are aware that the LFA test cannot detect positive samples between 30-35 Ct values, and when they are included in the group above 20 Ct values ​​in our study, it caused a lower sensitivity. However, as the power analysis of the study required this number of samples, we did not consider it ethical as all authors to evaluate the 30-35 Ct interval separately from this group. In addition, although we did not share the Ct intervals in the text in the first format, when you look at Figure-2, 3, and 4, information about the Ct value of the samples detected as positive and negative is shown. Each point in Figures 2, 3, and 4 represents a sample. The horizontal axis of the graph gives the LFA test result (0; negative, 1; positive), and the vertical axis gives the rRT-PCR Ct value.

  15. We agree with the reviewer and added the following sentence to the last paragraph of the discussion as “However, the results of this test should be approached with extreme caution. Because, Panbio™ COVID-19/Flu A&B Rapid Panel test kit y is prone to produce false negatives for the higher Ct values, in case of the low viral loads for the detection of SARS-CoV-2, INF-A, and INF-B”.

  16. In the evaluation of the LFA test for the detection of three viruses, attention was paid to the homogeneous distribution of each Ct value within the group. In addition, care was taken to avoid accumulation at low or high Ct values ​​in both groups. The range of Ct values ​​grouped in lines 185-186 in the conclusion section is specified.

  17. It has been edited.
  18. We fully agree with the referee's comments. “Lines 254-258: “The US Food and Drug Administration (FDA) emphasizes that a rapid antigen test should have at least 80% sensitivity and 98% specificity” we saw this statement in an article and overlooked the reference accuracy. For this reason, both the source and the expression have been removed from the article.

  19. Diagnostic performance studies for combined or Pan-antigen tests are limited to the best of our knowledge. Because; Considering that the study data of Widyasari et al. is valuable, we would like to present it in the discussion. As stated by the reviewer, the Ct intervals in our study should not be compared with the results of Widyasari et.al., because they are inconsistent with our studied Ct range.  In this direction, expressions related to the comparison of our results with the results of Widyasari et.al. were removed from our text.

  20. Revisions were made in line with the referee's suggestions
  21. We agree with the reviewer and we revised this sentence as “In conclusion, this kit demonstrated high sensitivity to SARS-CoV-2 and IAV for viral loads be-low 20 Ct values, but the sensitivity values were not compatible with PCR positivity for lower viral loads over 20 Ct values. Rapid antigen tests may be preferred as a routine screening tool in communal environments, especially in symptomatic individuals, when diagnosing SARS-CoV-2, IAV, and IBV with high caution.”. 
  22. We have received an editing service from MDPI for the manuscript.

Round 2

Reviewer 3 Report

Authors did a good job revising the manuscript; several new additions, such as explanations on the Ct values and their outcomes with the LFA test are adequately described which added great value to the manuscript. The manuscript is now well read and offers some useful information in the field of diagnostics. 

I have one typing correction for the authors to consider: Line 360: "Low-er". 

Well done! Congratulations!